

# The relationship between mobile phone location sensor data and depressive symptom severity

Sohrab Saeb[1,2], Emily G. Lattie[1], Stephen M. Schueller[1], Konrad P. Kording[2] and David C. Mohr[1]

[1] Department of Preventive Medicine, Northwestern University, Chicago, IL, United States
[2] Rehabilitation Institute of Chicago, Department of Physical Medicine and Rehabilitation, Northwestern University, Chicago, IL, United States

Corresponding author
David C. Mohr,
d-mohr@northwestern.edu

## ABSTRACT

**Background.** Smartphones offer the hope that depression can be detected using passively collected data from the phone sensors. The aim of this study was to replicate and extend previous work using geographic location (GPS) sensors to identify depressive symptom severity.

**Methods.** We used a dataset collected from 48 college students over a 10-week period, which included GPS phone sensor data and the Patient Health Questionnaire 9-item (PHQ-9) to evaluate depressive symptom severity at baseline and end-of-study. GPS features were calculated over the entire study, for weekdays and weekends, and in 2-week blocks.

**Results.** The results of this study replicated our previous findings that a number of GPS features, including location variance, entropy, and circadian movement, were significantly correlated with PHQ-9 scores ($r$'s ranging from $-0.43$ to $-0.46$, $p$-values $< .05$). We also found that these relationships were stronger when GPS features were calculated from weekend, compared to weekday, data. Although the correlation between baseline PHQ-9 scores with 2-week GPS features diminished as we moved further from baseline, correlations with the end-of-study scores remained significant regardless of the time point used to calculate the features.

**Discussion.** Our findings were consistent with past research demonstrating that GPS features may be an important and reliable predictor of depressive symptom severity. The varying strength of these relationships on weekends and weekdays suggests the role of weekend/weekday as a moderating variable. The finding that GPS features predict depressive symptom severity up to 10 weeks prior to assessment suggests that GPS features may have the potential as early warning signals of depression.

# INTRODUCTION

Depression is common and debilitating, taking an enormous toll in terms of cost, morbidity, and mortality (*Ferrari et al., 2013*; *Greenberg et al., 2015*). The 12-month prevalence of major depressive disorder among adults in the US is 6.9% (*Kessler et al., 2005*), and an additional 2–5% have subsyndromal symptoms that warrant treatment

(*Ayuso-Mateos et al., 2010*; *Kessler et al., 1997*). While treatments can be effective, failure to identify depression is a major factor in population-level disability. The US Preventive Services Task Force recommends annual screening for depression (*Siu et al., 2016*), and many argue that it should be far more frequent for at-risk patients (*Reynolds 3rd & Frank, 2016*). Although screening alone has little effect on the management of depression by clinicians (*Gilbody, Sheldon & House, 2008*), even annual screening does not occur regularly, and only 37% of individuals with depression receive treatment in the first year of an episode. In fact, the median time to treatment in the US is 8 years (*Wang et al., 2005*). Furthermore, the Affordable Care Act and the Mental Health Parity Act not only require access to mental health services, but also the measurement of the quality of those services with symptom and functional outcomes (*Basch, Torda & Adams, 2013*). Thus, the healthcare system relies almost entirely on people with depression to present themselves and accurately report their symptoms, both to initiate treatment and for follow up. This is despite the fact that depressed individuals commonly experience loss of motivation, stigmatization, and a sense of hopelessness and helplessness (*Mohr et al., 2010*). Therefore, identification of patients experiencing treatable levels of depression in a timely manner is a substantial failure point in the healthcare system.

The mobile phone is arguably the most ubiquitous personal sensing device, with nearly two-thirds (64%) of adults in the United States owning smartphones (*Pew Research Center, 2005*). These devices contain a growing complement of sensors which can provide data directly from the context of people's lives, and algorithms can translate phone sensor data into indicators of behavioral, social, and psychological targets. For example, Android provides activity status (walking, running, cycling, in vehicle, etc.) using the phone sensors. A growing body of research has demonstrated the potential of mobile phone sensors to detect a variety of behaviors related to depression, such as activity, sleep, and social interactions (*Abdullah et al., 2014*; *Grunerbl et al., 2015*; *Kwapisz, Weiss & Moore, 2011*; *Min et al., 2013*; *Shoaib et al., 2014*; *Thomée, Härenstam & Hagberg, 2011*; *Wiese et al., 2015*). Thus, a mobile phone sensing platform that detects depression could transform the management of depression, allowing for continuous and ubiquitous diagnostics for at risk populations.

Detection of depression using phone sensors, however, is a much more difficult task than detecting behaviors that are observable and more proximal to the sensor data such as physical activity, sleep/wake patterns, and social interaction. Depression has a variety of symptoms and experiences that can include depressed mood, loss of interest, sleep disturbances, lack of energy, appetite disruption, trouble concentrating, psychomotor disturbances, feelings of hopelessness, guilt and worthlessness, social withdrawal, irritability, and suicidal thoughts (*American Psychiatric Association, 2013*). While some of these symptoms might be captured through smartphone sensors, many cannot. Thus, there is no single set of sensors that is reliably and consistently sensitive to depression.

Detection of depression will likely require the development of *features* that translate mobile phone sensor data into behavioral targets that may be relevant to depression. We have recently done this with GPS data (*Saeb et al., 2015a*). Using GPS data collected from 28 participants over two weeks, we used a clustering algorithm to find each participant's favorite location. We then used these locations to calculate a number of features, including

*entropy*, or the variability in the time spent across favorite locations, and *circadian movement*, or the periodicity of movement between those locations. We found a number of features to be significantly correlated with depressive symptoms severity. Therefore, while it is difficult to relate raw sensor data or their basic statistics directly to depression, their features are more likely to indicate the presence of depressive symptoms.

The purpose of this study was to replicate and extend our previous findings using a separate dataset. Replication is a critical step in the area of behavioral sensing. While exploration is essential to discovery, it is frequently done in small datasets. This can lead to spurious findings, as has recently been seen in the failure to replicate a widely cited paper (*Likamwa et al., 2013*) on the use of phone sensor data to detect mood (*Ruwaard et al., 2016*). Thus, the first aim was to replicate our previous findings on the relationship between GPS features and depression. Our second aim was to extend these findings by exploring those relationships in workdays and non-workdays. Movement on workdays is likely determined to some degree by social roles and expectations, while movement on non-workdays is likely less determined by external demands and more by the individual's motivational state. We therefore hypothesized that GPS features would be more consistently related to depression on non-workdays, operationalized as weekends, than on workdays, operationalized as weekdays. The third aim was to explore the temporal directionality in the relationship between depression and GPS features.

## METHODS

### Data

We used a dataset from a study that was designed and conducted by researchers at Dartmouth College (*Wang et al., 2014*). This dataset, known as StudentLife, was collected from 48 students of a computer science class for a duration of 10 weeks. The students consisted of 38 males and 10 females. Two of them were first-year, 14 second-year, six third-year, and eight fourth-year Bachelor's students. There were also 13 first-year and one second-year Master's student, and three PhD students. Participants were racially diverse, with 23 Caucasians, 23 Asians, and two African-Americans. There was no inclusion or exclusion criteria based on the volunteers' mental health states. The authors of this paper had no involvement in the design of this study or the collection of these data. The study was approved by the Dartmouth College Institutional Review Board, and the study participants signed a consent form after being detailed about the type of data collected from their phones.

Data was collected using the StudentLife app, installed on Android devices. The students who did not own Android devices were provided with Google Nexus 4s phones for the period of the study. The StudentLife dataset consisted of a number of variables that were consistent with our previous study (*Saeb et al., 2015a*), including continuous phone GPS data and the Patient Health Questionnaire 9-item (PHQ-9) (*Kroenke, Spitzer & Williams, 2001*), a self-report measure of depression symptom severity. GPS data was collected continuously over 10 weeks, with a frequency of once in every 5 min. The PHQ-9 was administered at baseline and at the end of the study (follow-up).

Phone usage data in StudentLife was collected in a manner that was very different and not comparable to the data in our previous study (*Saeb et al., 2015a*): while we had logged

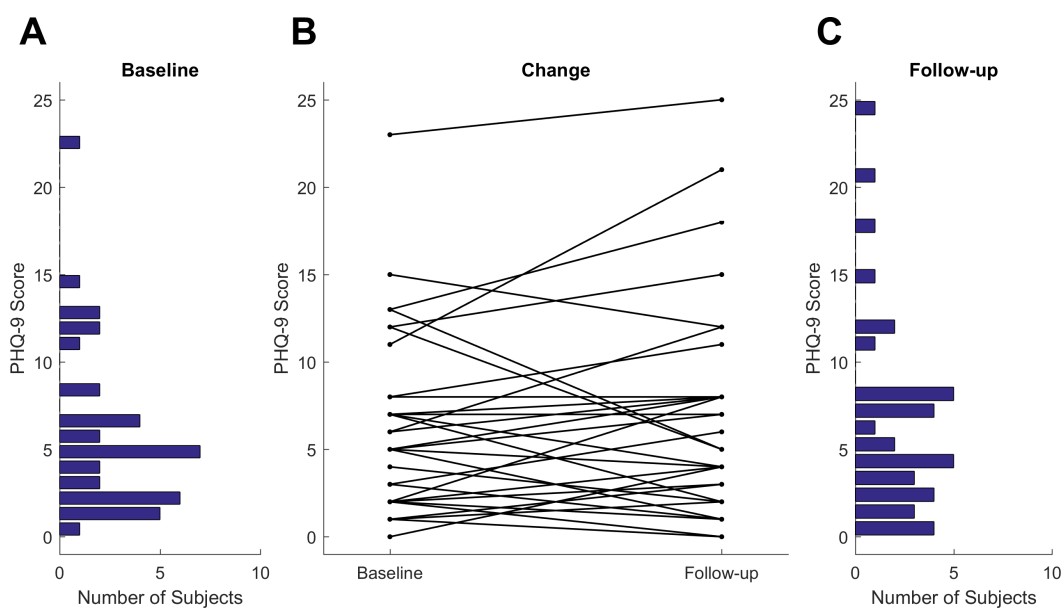

**Figure 1** **PHQ-9 score distribution at baseline (A) and the end of study follow-up (C). (B) shows the change from baseline to follow-up.** Each line represents one participant.

every screen on/off event, they only logged these events when the phone lock duration was longer than 1 h. We therefore focused our replication only on GPS.

The PHQ-9 scores were diverse across the participants and changed noticeably from baseline to follow-up, as shown in Fig. 1. For 20 participants, the PHQ-9 scores decreased from baseline to follow-up, while for 16 they increased. However, only seven participants had a change which was clinically meaningful, meaning that the absolute value of the change was greater than or equal to 5.

## Location features

We first extracted the spatial and temporal properties, or features, of the GPS sensor values. We used the same eight features that we had used in our previous study (*Saeb et al., 2015a*), plus three new exploratory features: speed mean, speed variance, and raw entropy. Table 1 gives a brief description of all 11 features.

Each location feature was calculated in three different ways:

### 10-week features

We extracted these features from all GPS data collected during the entire 10 weeks of the study.

### Weekday/weekend features

We extracted these features separately for workdays and non-workdays. Because we did not have the participants' school or work schedules, we operationalized these as *weekdays* (Monday through Friday) and *weekends* (Saturday and Sunday). Then, we calculated weekday and weekend features separately from each of these two sets.

**Table 1  Features used in this study and their definitions.** Features indicated with stars ($*$) are replicated from our previous study (*Saeb et al., 2015a*).

| Feature | Definition |
|---|---|
| Location variance* | Combined variance of latitude and longitude values: $$\text{Location variance} = \log(\sigma_{\text{lat}}^2 + \sigma_{\text{long}}^2),$$ where $\sigma_{\text{lat}}^2$ and $\sigma_{\text{long}}^2$ are the variance of latitude and longitude, respectively. |
| Circadian movement* | First, we used the least-squares spectral analysis (*Press, 2007*) to obtain the spectrum of the GPS signals. Then, we calculated the amount of energy that fell into the frequency bins within a $24 \pm 0.5$ h period, in the following way: $$E = \frac{1}{i_u - i_L} \sum_{i=i_L}^{i_u} \text{psd}(f_i),$$ where $\text{psd}(f_i)$ denotes the power spectral density at frequency bin $f_i$, and $i_L$ and $i_U$ represent the lower and the upper bounds of the frequency range of interest, corresponding to 24.5 and 23.5 h periods respectively. We calculated $E$ separately for longitude and latitude, and obtained the total circadian movement as: $$CM = \log(E_{\text{lat}} + E_{\text{long}})$$ |
| Speed mean | Mean of the instantaneous speed obtained at each GPS data point. The instantaneous speed (degrees/sec) was calculated as the change in latitude and longitude values over time in the following way: $$V_i = \sqrt{\left(\frac{\text{lat}_i - \text{lat}_{i-1}}{t_i - t_{i-1}}\right)^2 + \left(\frac{\text{long}_i - \text{long}_{i-1}}{t_i - t_{i-1}}\right)^2},$$ where $\text{lat}_i$, $\text{long}_i$, and $t_i$ are latitude, longitude, and time at sample $i$. |
| Speed variance | Variance of the instantaneous speed. |
| Total distance* | Total geographic displacement, as: $$\text{Total distance} = \sum_i \sqrt{(\text{lat}_i - \text{lat}_{i-1})^2 + (\text{long}_i - \text{long}_{i-1})^2},$$ where $\text{lat}_i$ and $\text{long}_i$ show latitude and longitude values at sample $i$. |
| Number of clusters* | Number of location clusters found by the adaptive $k$-means algorithm (*Saeb et al., 2015a.*). |
| Entropy* | Information theoretical entropy (*Shannon, 1997*), which measured how each participant's time was distributed over different location clusters: $$\text{Entropy} = -\sum_{i=1}^{N} p_i \log(p_i),$$ where $p_i$ is the percentage of time spent at location $i$, and $N$ is the total number of location clusters. |

**Table 1** (*continued*)

| Feature | Definition |
|---|---|
| Normalized entropy* | Entropy normalized by the number of location clusters ($N$): <br><br> $$\text{Normalized entropy} = \frac{\text{Entropy}}{\log(N)}$$ |
| Raw entropy | Same as entropy, with $p_i$ representing the number of data points in each latitude or longitude bin before clustering. A total number of $N = 10$ bins were used. The total raw entropy was defined as the sum of latitude and longitude raw entropies. |
| Home stay* | Percentage of time spent at home. |
| Transition time* | Percentage of time spent in transit, such as in a car or on bike. |

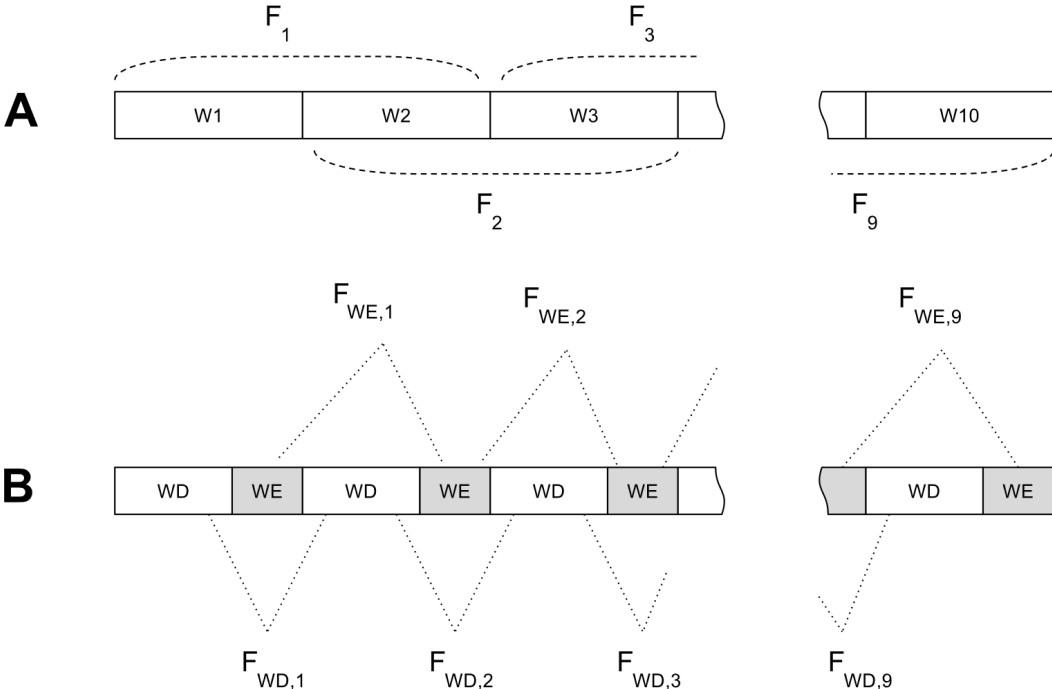

**Figure 2  Feature extraction procedure for 2-week features.** (A) The first set of features ($F_1$ to $F_9$) were extracted from 2-week blocks of sensor data that had an overlap of 1 week. (B) The second set of features were extracted after each week of data was split into weekday (Monday to Friday) and weekend (Saturday and Sunday). Weekday features ($F_{WD,1}$ to $F_{WD,9}$) were extracted from the weekday part and weekend features ($F_{WE,1}$ to $F_{WE,9}$) from the weekend part of each 2-week block.

### 2-week features

To support analyses examining directionality of the correlations, we calculated these features from 2-week-long blocks of GPS data with an overlap of 1 week (Fig. 2A). The 2-week period was selected as a block of time because a diagnosis of depression requires the presence of symptoms more days than not for two weeks. Therefore, we obtained nine sets of features corresponding to 10 weeks. Similar to 10-week features, we split these features into weekday and weekend features (Fig. 2B).

**Table 2 Linear correlation coefficients ($r$) between individual 10-week features and PHQ-9 scores, and their 95% confidence intervals.** Features indicated with stars (∗) are replicated from our previous study (*Saeb et al., 2015a.*). Bold values indicate significant correlations.

| Feature | Baseline ($n = 46$) | Follow-up ($n = 38$) | Change ($n = 38$) |
|---|---|---|---|
| Location variance* | −0.29 ± 0.008 | **−0.43 ± 0.007** | −0.34 ± 0.008 |
| Circadian movement* | −0.34 ± 0.006 | **−0.48 ± 0.006** | −0.33 ± 0.009 |
| Speed mean | −0.03 ± 0.007 | −0.06 ± 0.005 | 0.04 ± 0.008 |
| Speed variance | −0.07 ± 0.007 | −0.06 ± 0.005 | 0.06 ±0.007 |
| Total distance* | −0.23 ± 0.004 | −0.18 ± 0.006 | −0.03 ± 0.006 |
| Number of clusters* | **−0.38 ± 0.005** | **−0.44 ± 0.004** | −0.24 ± 0.007 |
| Entropy* | −0.31 ± 0.007 | **−0.46 ± 0.005** | −0.28 ± 0.008 |
| Normalized entropy* | −0.26 ± 0.007 | **−0.44 ± 0.005** | −0.30 ± 0.009 |
| Raw entropy | 0.17 ± 0.009 | 0.22 ± 0.008 | 0.15 ± 0.010 |
| Home stay* | 0.22 ± 0.008 | **0.43 ± 0.005** | 0.30 ± 0.009 |
| Transition time* | −0.30 ± 0.006 | −0.32 ± 0.005 | −0.12 ± 0.009 |

## Data analysis

We evaluated the relationship between each set of features (10-week and 2-week, each for all days, weekends, or weekdays) and depressive symptoms severity as measured by the PHQ-9. We used linear correlation coefficient ($r$) and considered $p < 0.05$ as the significance level. In order to reduce the possibility that results were generated by chance, we created 1,000 bootstrap subsamples (*Efron & Tibshirani, 1993*) to estimate these correlation coefficients and their 95% confidence intervals (CIs). We only considered those coefficients significant for which the 95% CI of their associated $p$-values fell below 0.05. Correlation analysis was conducted for baseline and follow-up PHQ-9 scores separately. In addition, although there were only seven participants who had clinically meaningful change ($\geq 5$) in their PHQ-9 scores, we also performed a correlation analysis with the change in PHQ-9 in order to see if any of the score changes, clinically meaningful or not, were related to GPS features.

## RESULTS

### 10-week features

The results of the correlation analysis between 10-week location features and depression scores are shown in Table 2. Results reaching the significance level (see 'Data Analysis') are shown in bold. Particularly, location variance, circadian movement, entropy, home stay, and number of clusters all had absolute correlation coefficients $|r| \geq 0.4$ with the follow-up PHQ-9 scores. Correlations between these features and baseline scores were weaker, and none of them were statistically significant. Overall, these results were consistent with our previous findings (*Saeb et al., 2015a*).

### Weekday/weekend features

The results of the correlations between GPS features and depression by weekday and weekend are shown in Table 3. All of those 10-week features that were significantly related to PHQ-9 scores (see Table 2) were also significant when calculated from weekends, whereas

**Table 3  Linear correlation coefficients (*r*) between individual weekend and weekday features and PHQ-9 scores, and their 95% confidence intervals.** Bold values indicate significant correlations (see 'Data Analysis').

| Feature | Weekday | | | Weekend | | |
|---|---|---|---|---|---|---|
| | Baseline (*n* = 46) | Follow-up (*n* = 38) | Change (*n* = 38) | Baseline (*n* = 46) | Follow-up (*n* = 38) | Change (*n* = 38) |
| Location variance | −0.15 ± 0.008 | −0.20 ± 0.008 | −0.22 ± 0.009 | −0.31 ± 0.008 | **−0.47 ±0.007** | −0.39 ± 0.008 |
| Circadian movement | −0.22 ± 0.007 | −0.28 ± 0.008 | −0.25 ± 0.009 | −0.35 ± 0.007 | **−0.51 ±0.006** | −0.36 ± 0.008 |
| Speed mean | −0.00 ± 0.008 | −0.06 ± 0.005 | 0.03 ± 0.008 | −0.13 ± 0.005 | −0.06 ± 0.006 | 0.05 ± 0.009 |
| Speed variance | −0.05 ± 0.008 | −0.07 ± 0.005 | 0.02 ± 0.007 | −0.13 ± 0.004 | −0.05 ± 0.006 | 0.10 ± 0.008 |
| Total distance | −0.20 ± 0.004 | −0.15 ± 0.005 | −0.01 ± 0.006 | −0.25 ± 0.004 | −0.20 ± 0.005 | −0.03 ± 0.006 |
| Number of clusters | −0.19 ± 0.006 | −0.25 ± 0.005 | −0.14 ± 0.008 | −0.34 ± 0.006 | **−0.46 ±0.004** | −0.32 ± 0.007 |
| Entropy | −0.21 ± 0.007 | −0.34 ± 0.006 | −0.20 ± 0.009 | −0.30 ± 0.008 | **−0.55 ±0.004** | −0.38 ± 0.008 |
| Normalized entropy | −0.21 ± 0.008 | **−0.39 ± 0.006** | −0.24 ± 0.009 | −0.28 ± 0.008 | **−0.54 ± 0.004** | −0.41 ± 0.009 |
| Raw entropy | 0.05 ± 0.008 | −0.04 ± 0.008 | 0.01 ± 0.010 | 0.04 ± 0.008 | −0.01 ± 0.008 | 0.03 ± 0.009 |
| Home stay | 0.19 ± 0.008 | 0.37 ± 0.006 | 0.23 ± 0.009 | 0.23 ± 0.007 | **0.50 ± 0.004** | 0.35 ± 0.008 |
| Transition time | −0.27 ± 0.006 | −0.29 ± 0.006 | −0.14 ± 0.010 | −0.36 ± 0.006 | −0.32 ± 0.008 | −0.06 ± 0.009 |

only normalized entropy was significantly related to the scores as a weekday feature. The magnitude of the relationship between weekend features and PHQ-9 scores was larger than the magnitude of the relationship between 10-week features and PHQ-9 scores. However, given the small sample size, we were not adequately powered to test if these differences were significant.

## 2-week features

Finally, we examined how 2-week GPS features obtained at different times during the study correlated with baseline and follow-up depression scores. These analyses were performed only on those features that showed significant relationships with PHQ-9 scores in the previous section (see Tables 2 and 3). We ran these analyses separately for weekday and weekend features, calculated at each week.

Consistent with the results in Table 3, correlations of weekend features (Figs. 3C–3D) were generally stronger than weekday features (Figs. 3A–3B). Furthermore, correlations between baseline PHQ-9 scores and weekend location features were significant in the first weeks immediately following depression assessment, but quickly became non-significant and approached zero at the end of the 10 weeks (Fig. 3C). In contrast, correlations between weekend location features and the follow-up PHQ-9 scores generally remained significant regardless of the time point at which features were extracted (Fig. 3D). This was also true for three out of the six features extracted on weekdays (Fig. 3B), including homestay, entropy, and normalized entropy. Overall, while these results were consistent with correlation results in Table 3, they indicated that these correlations changed over time.

## DISCUSSION

In this study, we were able to replicate our previous findings of the relationship between severity of depressive symptoms and GPS location features, including location variance, entropy, and circadian movement (*Saeb et al., 2015a*; *Saeb et al., 2015b*), in the StudentLife

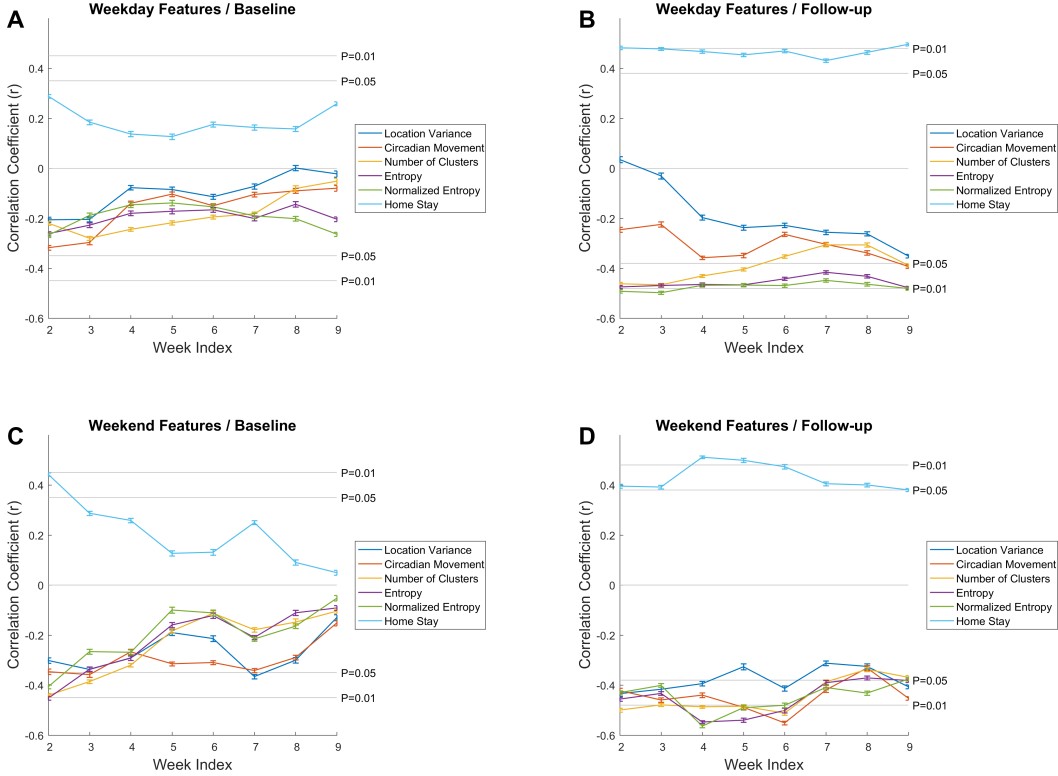

**Figure 3** **Mean temporal correlations between 2-week location features, calculated at different time points during the study, and baseline and follow-up PHQ-9 scores.** Error bars show the 95% confidence intervals. In (A–B), features were obtained from weekday data only, and in (C–D), they were extracted from weekend sensor data. For each 2-week feature set, week indices indicate when the 2-week period ended. Due to sparsity of data in week 10, we excluded it from this analysis.

dataset (*Wang et al., 2014*). This is particularly notable as the StudentLife dataset was collected completely independently of this research group, using a very different sample that consisted of college students in a small town as opposed to the general population in a large city (Chicago) from our first study. Furthermore, subsequent to our initial publication, a third independent group, Canzian and Musolesi, developed very similar location features and had similar findings (*Canzian & Musolesi, 2015*). Together, these findings provide strong evidence that features of GPS location such as the number of places a person goes, how time is spent in these locations, and the circadian rhythm of movement through geographic space, are strongly related to the severity of depressive symptoms.

Replication in the field of psychological science is important given how frequently replication studies fail. In fact, a recent meta-study showed that only one-third to one-half of experimental results reported in psychological science were replicated in later studies (*Open Science, 2015*). This is particularly crucial in the field of behavioral sensing using smartphones since studies in this field frequently use small samples, are usually not cross-validated, and failure to replicate has already been observed (*Ruwaard et al., 2016*). Thus, replication studies such as the present one can be seen as the studies that make findings real.

In addition to replication, we took this opportunity to extend previous findings in two important ways. First, we examined the relation of GPS features to depression on weekends and weekdays separately. We suspected that the relationship between GPS features and depression seen in our first study might reflect participants' motivational states. If true, we hypothesized that the signal would be stronger when the individual had more control over their own movement, such as on a non-workday, than when movement was in response to external demands, such as on workdays when a larger percentage of movement is typically determined by job or school-related demands. Indeed, we did see larger correlations on weekends between many of the GPS features and depression, than on weekdays.

The weekend vs. weekday finding illustrates the importance of considering a wide range of possible features or conditions that improve sensor-based detection of behaviors and mental health conditions. A large empirical and theoretical literature has focused on the complex environmental, situational, personality, and motivational factors that drive human behavior (*Fleeson, 2004*; *Mischel, 1969*; *Rauthmann et al., 2014*). Consideration of such complex interactions likely has considerable value in the development of sensor-based detection of behavior and mental health conditions. For example, simply looking at these GPS data, one might imagine a wide variety of features to improve the detection, such as urban versus rural, northern versus southern, and economically deprived versus wealthy locations, as well as season, weather, and climate. Such features can also be detected passively, and in combination they may improve the detection accuracy.

The second way we extended our previous findings was to examine the temporal relationship between GPS features and depression. The StudentLife study extended over 10 weeks, as compared to two weeks for our first study, making it a better dataset to explore temporal relationships. The relationship between depressive symptom severity and subsequent location features was significant for the first weeks, but then rapidly declined over time. This was most notable when using the weekend location features, which were generally stronger predictors. In contrast, the correlations between weekend location features immediately prior to the assessment of depression symptom severity were very similar to the correlation 8–10 weeks prior. This suggests that location features may be early warning detectors of depression.

It is also interesting to note that some location features, such as location variance and circadian movement, which were strongly predictive of depression 8–10 weeks prior when measured on weekends, showed no significant prediction when measured on weekdays. Nevertheless, weekday measurement did become stronger the closer their extraction occurred to the assessment. This may suggest that depression-related disruption in movement first manifests itself only when there are no social constraints (e.g., on non-workdays), but as the disruption in GPS features shifts to manifest itself on workdays, risk of subsequent depression becomes more immediate. Thus, the relationship between features over time may improve the prediction. For example, the weekend features 10 weeks prior, combined with weekday features at a later date may represent the shift of risk factors occurring in the absence of social demands to occurrence in the presence of social demands, thereby improving the prediction of depression. Such hypotheses can be tested in larger datasets.

There are a number of limitations that should be noted as well. First, the findings on weekend and weekday features and temporal relationships contain a large number of correlation analyses which introduce the possibility of alpha slippage, particularly for the exploratory analyses. Furthermore, we were not sufficiently powered to detect workday/non-workday differences in the relationship between GPS and depression, nor the differences in these relationships over time. Thus, these findings should be considered preliminary until replicated. In contrast, given the consistency across studies and populations, the number of analyses for the replication analyses is not a significant limitation for this study.

Second, the distinction between weekdays and weekend is not necessarily a defining feature for every person. Some schools might have schedules that do not fall cleanly into a 5-day work week and many people clearly work on weekends. In a previous analysis of the StudentLife dataset, for example, specific aspects about Dartmouth social behavior were used to explore typical "party" days which did not line up exactly with weekend/weekdays (Wang et al., 2015). Weekend/weekday was simply an operationalization for workday/non-workday. In a broader, non-student sample, this might be detected passively for most individuals by identifying their work location.

Finally, this study had a relatively small sample size. Our finding of the importance of workday/non-workday distinction highlights the likely importance of the variables that moderate the relationship between sensor data and mental health. There are likely many such variables, such as climate, age, and urban/rural locations, to name a few. Therefore, the development of effective behavioral sensing prediction models for mental health will likely require a much larger sample size. Furthermore, behavioral sensing studies—including ours—require longer periods of time to capture the slow changes in the disease state and in the behavioral features. Most studies of behavioral sensing in depression only have 2 or 3 assessment points for each individual, which makes the dataset underpowered for investigating the longitudinal relationship between sensor data—which has much higher frequency—and depression. Longer studies with more frequent assessments of depressive symptoms would allow researchers to more accurately model and predict these changes, both across the population and within each individuals.

In conclusion, our study supports the potential of smartphone sensor technology in providing biomarkers of depression in daily life. However, even with strong biomarkers, it remains a challenge to develop a system that can passively detect depression with low false positive and false negative rates. For this potential to be realized, well-designed studies with large numbers of participants over longer periods of time will be required. Such properly designed studies have the potential to transform mental health care, allowing for objective, ubiquitous, sensor-based evaluations that require little to no ongoing effort to estimate the risk of depression.

## ACKNOWLEDGEMENTS

The authors would like to thank Dr. Andrew T. Campbell for helpful discussions on the StudentLife dataset and for providing information on Dartmouth College Institutional Review Board approval.

### Funding

This study was funded by research grant P20 MH090318 from the US National Institute of Mental Health. Author SMS was supported by a grant from the National Institute of Mental Health K08 MH 102336. The funders had no role in study design, data collection and analysis, decision to publish, or preparation of the manuscript.

### Grant Disclosures

The following grant information was disclosed by the authors:
US National Institute of Mental Health: P20 MH090318.
National Institute of Mental Health: K08 MH 102336.

### Competing Interests

The authors declare there are no competing interests.

### Author Contributions

- Sohrab Saeb analyzed the data, contributed reagents/materials/analysis tools, wrote the paper, prepared figures and/or tables, reviewed drafts of the paper.
- Emily G. Lattie and Stephen M. Schueller wrote the paper, reviewed drafts of the paper.
- Konrad P. Kording and David C. Mohr contributed reagents/materials/analysis tools, wrote the paper, reviewed drafts of the paper.

### Human Ethics

The following information was supplied relating to ethical approvals (i.e., approving body and any reference numbers):

The original study was conducted by researchers at Dartmouth College, where the data was collected. The original study was approved by the Institutional Review Board at Dartmouth College. The authors of this paper did not have any role in the original study, and the secondary analysis presented in this paper did not need any IRB approval.

### Data Availability

The raw data has been supplied as a Supplemental Information, and the GPS location sensor data is publicly available at:

http://studentlife.cs.dartmouth.edu/dataset/dataset.tar.bz2.

### Supplemental Information

Supplemental information for this article can be found online at http://dx.doi.org/10.7717/peerj.2537#supplemental-information.

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
