# Peer review of "The relationship between mobile phone location sensor data and depressive symptom severity"

_PeerJ, doi:10.7717/peerj.2537_

## Round 0.1 · original submission · Major Revisions

The reviewers are quite positive about your paper, but have a number of improvements to suggest.

·

Basic reporting

No specific comments - minor corrections listed below.

Experimental design

No comments.

Validity of the findings

No specific comments - minor issues with discussion points listed below.

Additional comments

This paper reports on an interesting approach to identifying location-based correlates of depression scores. The paper is generally clear and discussion is appropriate. The examination of weekend vs weekday differences appears to be a useful avenue for differentiating location sensor data. This differentiation may have broader implications for how depression manifests in regard to activity levels, and additional discussion of these potential clinical implications might further strengthen the paper. I have a few additional comments.

1. Line 33: “Thus, our healthcare system...” – is this referring to the US healthcare system? Or can this be said of healthcare systems more broadly?
2. Line 68: Word missing: “...on the use [of] phone sensor data...”
3. The results around Figure 3 are confusing, as the text indicates that A/B are weekend features, whereas the Figure indicates these are weekday features. Does A/C represent initial depression scores and B/D final depression scores?
4. The use of tense is sometimes inconsistent – past tense should be used to describe the study and its findings as it has been completed. e.g., "the correlations between weekend location features immediately prior to the assessment of depression symptom severity is very similar to the correlation 8-10 weeks prior" - "is" should be "were" (and "correlation" should be "correlations")
5. Line 249: “differences in these relationships overtime” should be “over time”.
6. Added nuance should be given to the concept that the solution to increasing treatment rates for depression is regular screening. There is evidence that screening programs alone do little to change outcomes in depression (see, e.g., Gilbody et al 2008, Canadian Medical Association Journal, 178, 997-1003). Furthermore, there are many reasons beyond recognition/detection that people do not seek help for depression.
7. Line 86-87: “The authors of this paper had no involvement in the design of this study or the collection of these data.” – Why were the individuals who designed the study and collected the data not included as authors of the paper? Were they invited?
8. Line 173-175: “Overall, these results imply that GPS location features may have the potential to predict depressive states many weeks in advance.” – This sentence should be in the discussion. Moreover, these results may be overstated, as there are alternative explanations consistent with these results, e.g., depressive symptoms predict location features [depression symptoms were not assessed contiguously (i.e., every week) with location features].
9. Some discussion is given to longitudinal relationships. The authors might also consider how intra-individual changes might also predict mental health outcomes.
10. Further consideration of the barriers to detecting depression using sensor data should be considered in the discussion. Even with collection of multiple markers that are highly correlated with depression scores, it would be challenging to create a passive indicator of depression for the general population with low false positive/false negative rates. A lot of work remains to be done.

Reviewer 2 ·

Basic reporting

This is a well written, self-contained manuscript.

Experimental design

This is a replication study to validate previously derived features which relate to depression severity. The authors clearly articulate the importance of this work.

Validity of the findings

The analysis seems robust, and the limitations are clearly stated in the discussion.

My only query would be around the variability in the PHQ-9 scores at an individual level. Most participants' scores didn't change much, so could the GPS data track changes?

Additional comments

1. In the Methods/Data section, could the authors clarify which handsets the StudentLife data was collected from (e.g. Android and/or iOS).
2. In the Methods/Data section (line 83), it might be useful for an international audience to clarify the categories of freshman/junior/sophomore/senior.
3. Line 111 “participant’s” should be “participants’”
4. Table 1: in the definition of Circadian Movement, it looks like i2-i1 = 1 … is E simply psd(f1) + psd(f2)?
5. Table 1: was Speed Mean provided with the GPS co-ordinates by the relevant API, or was this subsequently derived as the change in position over time?
6. Table 1: are additional brackets needed in the Total Distance equation to indicate the sqrt function applies to the sum of both squares?
7. Suggest Table 2 and Table 3 are reported in the same order as Table 1, for clarity
8. Lines 166-167 indicate Fig 3A/B are weekend and Fig 3C/D are weekday. The captions in Figure 3 indicate the opposite? Also, please clarify if Fig 3A/C are correlations with the baseline data, and Fig 3B/D with the follow-up?

---

## Round 0.2 · Minor Revisions

Please consider the following minor changes:
1. Line 46. Make clear what country or countries the 68% figure applies to.
2. Line 67. "We the used" should be "We then used".
3. Line 124. "Figrue" is misspelled.
4. Lines 124-125. Either change "scores" to "score" or "it" to "they" to give agreement.
5. Line 158 "participants" should be "participants'".

---

## Round 0.3 · accepted · Accept

Thank you for the minor revisions. The paper is now suitable for publication.